# Multiple channelling single-photon emission with scattering holography designed metasurfaces

Danylo Komisar [1] ✉, Shailesh Kumar [1] ✉, Yinhui Kan [1], Chao Meng [1], Liudmila F. Kulikova[2], Valery A. Davydov[2], Viatcheslav N. Agafonov [3] & Sergey I. Bozhevolnyi [1,4]

Channelling single-photon emission in multiple well-defined directions and simultaneously controlling its polarization characteristics is highly desirable for numerous quantum technology applications. We show that this can be achieved by using quantum emitters (QEs) nonradiatively coupled to surface plasmon polaritons (SPPs), which are scattered into outgoing free-propagating waves by appropriately designed metasurfaces. The QE-coupled metasurface design is based on the scattering holography approach with radially diverging SPPs as reference waves. Using holographic metasurfaces fabricated around nanodiamonds with single Ge vacancy centres, we experimentally demonstrate on-chip integrated efficient generation of two well-collimated single-photon beams propagating along different 15° off-normal directions with orthogonal linear polarizations.

Single-photon emission is an essential part in different quantum information technologies, including quantum communication and cryptography[1], sensing[2], and in fundamental studies of quantum phenomena[3,4]. Solid-state systems like single molecules[5,6], colour centers in diamonds[7–9], quantum dots[10–12], and defects in 2D materials[13] are reported to serve as robust single-photon sources[14,15]. Germanium vacancy (GeV) centers in diamonds have emerged as a promising candidate for use in single-photon experiments due to their brightness, high emission into zero-phonon line and ability to operate at room temperatures[16–22]. However, the spontaneous emission of quantum emitters located in free space is, in general, not well-directed and poorly polarized. Rapidly developing quantum technologies require a technological platform that would provide a reliable method for realizing, preferably on-chip, versatile and efficient single-photon sources with controlled emission directions and polarizations.

Generally, there are two different routes for manipulating photon emission from QEs, viz. far- and near-field approaches. In the far-field configurations, the photon emission is manipulated using bulky optical components, like lenses, mirrors, and polarizers. In recent years, metalenses have been introduced to manipulate nonclassical light in a manner of their bulky counterparts, i.e., operating as external components that influence the radiation focusing, direction, and polarization after the quantum emission was generated[23–26]. In near-field configurations, QEs are nonradiatively coupled to metasurfaces, which can generate polarized and directional photon emission directly by shaping the out-of-plane scattering of surface (nonradiative) modes. This approach is apparently more efficient and compact by virtue of dispensing with the external components[26,27].

Following near-field approach, various plasmonic antennas have been utilized for engineering single-photon properties and have been used for generation of radially polarized[28], circularly polarized[29] and vortex[27] beams, polarization encoded multichannel emission[30], cavity-enhanced emission[31] and beam steering[32]. Many of these plasmonic antennas represent optical metasurfaces, i.e., dielectric ridge structures fabricated around QEs, that are utilized to scatter QE-excited SPPs into outgoing free-propagating waves, enabling thereby control over the QE emission into free space[27–29]. These metasurfaces feature, however, a fundamental limitation, which is associated with the

[1]Centre for Nano Optics, University of Southern Denmark, DK-5230 Odense M, Denmark. [2]L.F. Vereshchagin Institute for High Pressure Physics, Russian Academy of Sciences, Troitsk, Moscow 142190, Russia. [3]GREMAN, CNRS, UMR 7347, INSA CVL, Université de Tours, 37200 Tours, France. [4]Danish Institute for Advanced Study, University of Southern Denmark, DK-5230 Odense M, Denmark. ✉e-mail: dak@mci.sdu.dk; shku@mci.sdu.dk

isotropic nature of SPP scattering by continuous antenna ridges and which results in an inherent inability to efficiently control the emission polarization, unless the radially polarized emission is targeted[28]. The diverging QE-excited SPP is radially polarized in the surface plane. This polarization is then projected onto the polarization of the outgoing free-propagating waves produced by a metasurface. This allows for control over the polarization of the outcoupled beam. For example, the generation of the left and right circularly polarized beams with specific angular optical momenta[27,33]. The metasurface design can be described within the framework of the scattering-holography approach[33] that allows one to accurately design metasurfaces for single-channel photon emission into well-defined arbitrary, including off-normal, directions. The approach developed in this work makes one step further by splitting orthogonal polarizations and sending collimated beams in two different directions.

In this article, we report on the design, fabrication and characterization of hybrid QE-SPP coupled metasurfaces that enable generation of two well-collimated (divergences < 6.5°) single-photon beams propagating along different 15° off-normal directions and featuring orthogonal linear polarizations. We find that the external quantum efficiency of metasurface-coupled QE emitting photons into the two beams can exceed 80%, depending on the metasurface realization. Highly compact and efficient planar photonic components demonstrated in our work represent essentially single-photon sources, collimators and polarizing beam splitters integrated together in on-chip devices, showcasing thereby the potential of QE-SPP coupled holographic metasurfaces for a wide range of quantum technology applications.

## Results

### Performance of plasmonic metasurface

The operation principle and performance of the designed QE-SPP coupled metasurface is illustrated in Fig. 1. The fabricated metasurface consists of a single GeV center in a nanodiamond (ND) with an emission peak at 602 nm placed atop an optically thick silver (Ag) film with a thin $SiO_2$ overlayer and surrounded by dielectric outcoupling ridges. When pumped with a 532 nm laser, the GeV-ND center, which is selected to have the radiative transition dipole oriented perpendicular to the sample surface, excites radially diverging SPPs that propagate along the Ag–$SiO_2$ interface. The SPP excitation efficiency depends on the

configuration material, optical and geometrical parameters, influencing the loss through nonradiative and radiative channels, and can exceed 85%[28,34]. These SPPs are then out-of-plane scattered by an array of hydrogen silsesquioxane (HSQ) ridges into outgoing free-propagating waves in the form of two single-photon Gaussian beams, featuring orthogonal linear polarizations and propagating along different directions. The inevitable absorption of SPPs occurring during their propagation (before being out-of-plane scattered) can be minimized by using high-refractive index ridges to increase the scattering efficiency and high-quality metal films to reduce the ohmic losses. For example, in the case of generation of radially polarized single photons in a similar metasurface configuration with the bullseye ridge pattern, the external quantum efficiency can reach 95%[28]. The external quantum efficiency is defined as a fraction of the total dipole power radiated into outgoing waves, propagating within a 64° cone (collected by an NA = 0.9 objective). For the considered configuration, the corresponding external quantum efficiency is evaluated at the level of 60% (see Supplementary Section S2.3 for details).

### Holography approach to metasurface design

The QE-SPP coupled metasurface design is based on the scattering-holography approach with the QE-excited radially diverging SPPs serving as reference waves[33], which we further extend to the case of photon emission into multiple different directions. This is achieved by multiplexing holograms, i.e., the intensity interference patterns, computed when using the in-plane field distributions of signal beams with desired sizes, polarizations and propagation directions (Supplementary Figs. S1 and S2). For clarity, we consider only the in-plane electric field components, as they are the ones responsible for scattering in the directions close to normal to the surface (Supplementary Fig. S3). In our case, the electric fields of the signal waves are represented by their in-plane field components $E_x$ and $E_y$ for either X- or Y-polarized beams propagating along different 15° off-normal directions. Note that, due to the orthogonal polarizations of the two signal waves, the hologram can be computed in one step (Supplementary Fig. S4) instead of computing holograms separately with their subsequent multiplexing. Importantly, in this special case of orthogonal linear polarizations, two constituting holograms complement each other resulting in the efficient use of the available metasurface area (Fig. 2b, c). Generally speaking, the efficiency of isotropic holographic metasurfaces for producing linearly polarized free space emission cannot exceed 50%, because in-plane radially polarized SPPs contain equal amount of two orthogonal linear polarizations. For this reason, designing the QE-SPP coupled metasurface for generating two single-photon beams with orthogonal linear polarizations is attractive apart from realizing a useful single-photon functionality also from the viewpoint of maximizing the external quantum efficiency.

The practical design of the metasurface ridges builds upon the calculated smooth interference patterns, requiring their conversion into patterns of ridges of the same height but different width. Note that the presence of dielectric ridges influences the SPP propagation constant and has to be properly taken into account already at the stage of calculating the interference pattern[28,29,33]. To simplify the design procedure we adjust our design so that the ridge filling factor varies around 0.5, and use this filling factor and the ridge height of 200 nm to evaluate the SPP propagation constant for computing the interference patterns[28]. Importantly, when calculating a particular interference pattern we use radially increasing SPP field as reference wave. This is done to compensate for the radial decay of QE-excited SPP wave due to its divergence, absorption and out-of-plane scattering (Supplementary Fig. S5). This procedure results in widening of outer metasurface ridges and thus enhancing the SPP scattering. As a result, when the designed metasurface interacts with diverging and radially attenuating SPP waves, the out-of-plane scattered field reconstructs a predefined amplitude and polarization profile[33].

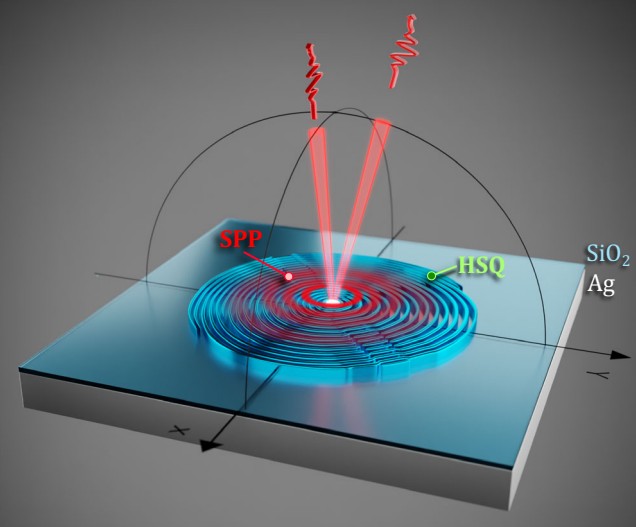

**Fig. 1 | Schematic of a QE-SPP coupled metasurface.** QE-excited radially diverging SPPs are scattered by metasurface dielectric ridges into outgoing free-propagating waves, producing two spatially separated single-photon beams with orthogonal linear polarizations.

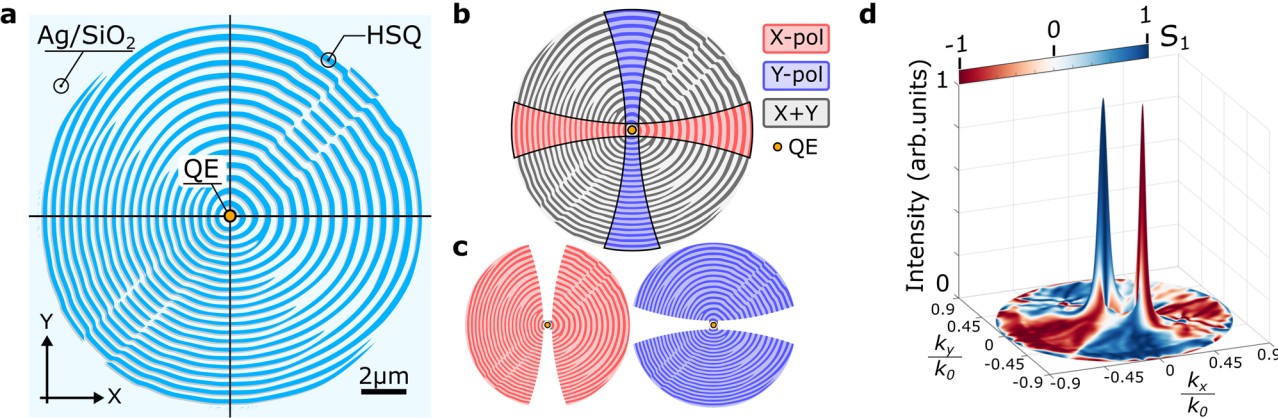

**Fig. 2 | Hologram metasurface design and operation. a** Optimized pattern of HSQ ridges composing the polarization-splitting metasurface. **b** Illustration of the operation principle behind generation of orthogonally polarized beams: red and blue coloured areas indicate metasurface domains responsible for generating predominantly X- and Y-polarized outgoing waves, respectively. Grey coloured domains contribute to both X- and Y-polarized beams. Metasurfaces designed for generating either X-(red) or Y-(blue) polarized beams are illustrated in (**c**).

**d** Calculated 3D angular distributions of the superimposed beam intensity shown by height and polarization Stokes parameter $S_1$ encoded by colour. The angular coordinates $k_x/k_0$ and $k_y/k_0$ are x- and y- wavevector projections on the image plane normalized to the free space wavevector. These simulations are conducted for the optimized metasurface configuration described further in the paper. The occurrence of blue and red peaks demonstrates spatial separation of two emission channels featuring orthogonal linear polarizations.

## Pattern optimization

The optimization of the holographic metasurface to be fabricated was conducted numerically at the wavelength of 602 nm corresponding to the zero-phonon line (ZPL) of the GeV center. In particular, we optimized the metasurface diameter along with the ridge width variations in order to maximize the external quantum efficiency and minimize the divergence of each beam, while preventing the fabrication of unnecessarily large metasurface structures. The metasurface size is determined by the SPP decay due to its absorption and scattering on the ridges. By considering these factors, we calculated an effective structure diameter of 17 μm (Supplementary Fig. S7). The resulting metasurface presented in Fig. 2(a) is expected to produce two well-separated (overlap < 13%) beams with the full angular width at half maximum (FWHM) of $\theta_{1/2} < 5.2°$ and the external quantum efficiency of 63%. At the same time, the polarization states of generated beams characterized by the Stokes parameter $S_1$, which is determined as $S_1 = (I_x - I_y)/(I_x + I_y)$ with $I_x(I_y)$ denoting the intensities of the linear X(Y) polarizations, are expected to be orthogonal with high degrees of linear polarizations (Fig. 2d). Finally, we assessed the robustness of the resulting metasurfaces design for eventual ND misalignments with respect to the metasurface ridges, and found that the ND displacements in the surface plane below 100 nm do not noticeably perturb the far-field emission pattern and thereby the metasurface performance (Supplementary Fig. S10).

## Metasurface fabrication verification

A scanning electron microscope (SEM) image of the fabricated GeV-ND-coupled metasurface with a diameter of 17 μm is shown in Fig. 3a, b. A typical fluorescence scan of the properly oriented GeV-ND, coupled to the metasurface and measured with a narrow-band spectral filter at 605 ± 8 nm, is shown in Fig. 3c. The doughnut-shaped pattern of fluorescence confirms that the considered ND contains a GeV center with a radiative dipole oriented perpendicularly to the sample surface, as required for maximizing SPP excitation and for proper operation of the metasurface designed for the dipole source normal to the surface.

## Single-photon source characterization

We characterize properly oriented GeV-NDs before and after the metasurface fabrication. Typical normalized emission spectra of a GeV-ND center obtained before and after its coupling to the metasurface

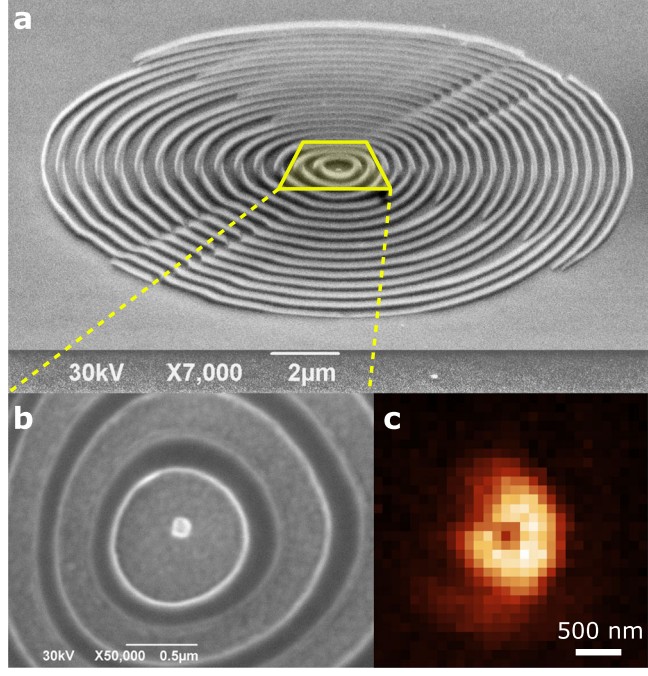

**Fig. 3 | Images of the fabricated metasurface and GeV-ND. a** Tilted SEM image of the holographic metasurface. **b** Zoom on the central area showing an ND with a GeV center properly positioned with respect to metasurface ridges. **c** Fluorescence scan image (3 × 3 μm²) showing emission of the GeV with out-of-plane dipole moment.

(Fig. 4a) feature a characteristic ZPL centered at 602 nm with 7 nm FWHM[35,36]. After the coupling to the metasurface in this particular case, the ZPL signal became enhanced by 20%, an enhancement which is related to the efficient SPP outcoupling by the metasurface ridges. The lifetime measured for the GeV-ND before and after fabrication of the metasurface are $\tau_{uc} = 16$ ns and $\tau_c = 19$ ns, respectively, corresponds to typical values[16,31,37]. An increase in lifetime after fabrication is attributed to a destructive feedback by metasurface ridges surrounding the GeV-ND asymmetrically (Fig. 3b). A single exponential model, $I = A_0 + Ae^{-t/\tau}$, is used for fitting the experimental data, where $\tau$ is a characteristic lifetime

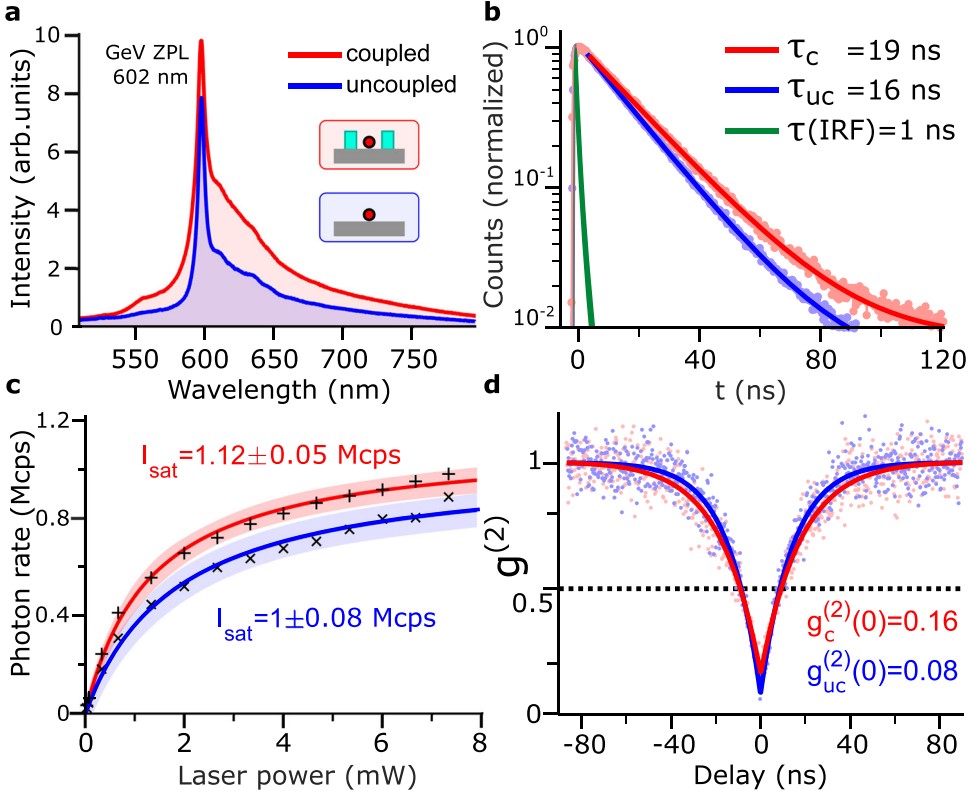

**Fig. 4 | Characterization of GeV-ND emission. a** Emission spectra of GeV-ND center characterized before (blue) and after (red) coupling to the metasurface featuring the resonant peak at 602 nm. **b** Lifetime measurements using a pulsed laser with 1 MHz rate show single-exponent emission decays along with the instrument response function (IRF). **c** Emission rate power-saturation dependencies. Error boundaries (shadow regions) represent 95% fitting confidence intervals. **d** Correlation data and a model fit to the $g^{(2)}$ correlation function. In figures (**b**–**d**) dots, plus signs and crosses represent raw data and solid lines represent fitting curves.

of the decay process with $A_0/A$ accounting for a background contribution (Fig. 4b).

To estimate the GeV-ND center brightness, we measured emission rates while increasing the excitation power and fitted the data to estimate the saturated emission rate and excitation power (Fig. 4c). The data were fitted to $I = I_{sat}[P/(P + P_{sat})]$, where $I$ and $P$ are fluorescence count rate and excitation power, respectively, whereas $I_{sat}$ and $P_{sat}$ are the fitting parameters. It is seen that the saturated emission rate before coupling to the metasurface, $I_{sat(uc)} = 1 \pm 0.08$ Mcps and $P_{sat(uc)} = 1.8 \pm 0.04$ mW, increased after the coupling to $I_{sat(c)} = 1.12 \pm 0.05$ Mcps and $P_{sat(c)} = 1.3 \pm 0.2$ mW. The increased single-photon emission rate observed is consistent with the increased ZPL signal (Fig. 4a) and related to the efficient SPP outcoupling by the metasurface that outweighs the influence of its destructive feedback.

Finally, we verified the single-photon nature of emission from the investigated GeV-ND by characterizing the autocorrelation of its emission and observing anti-bunching minima well below 0.5 both before, $g_{uc}^{(2)}(0) = 0.08$, and after, $g_c^{(2)}(0) = 0.16$, coupling to the metasurface (Fig. 4d). We attribute a slight increase in $g^{(2)}(0)$ to the background fluorescence of the HSQ traces around the GeV-ND.

**Polarization splitting and beams collimation**
Having confirmed the single-photon emission from the GeV-SPP coupled metasurface, we investigated angular distribution and polarization characteristics of the single-photon emission, observing in the Fourier plane two bright emission spots with orthogonal linear polarizations as expected (Fig. 5a). The spot locations are equally displaced from the normal to the surface by 15° as designed and also predicted by FDTD calculations (Fig. 5b), demonstrating good agreement between experimental and calculated angular distributions. The

generated beams are expected to be collimated better in the direction of the emission polarization due to a better coverage by the corresponding metasurface region (Fig. 2b, c), a feature that is ultimately related to the longitudinal orientation of the in-plane SPP polarization. Nevertheless, the experimentally measured angular FWHM of the generated beams for both polarizations was found within 6.5°, securing thereby sufficient angular separation of the emission channels with orthogonal linear polarizations. A small difference in experimental X and Y peak intensities (Fig. 5c) is believed to result from imperfection of the metasurface positioning, whose precision is evaluated being 50 nm. The experimentally realized degree of linear polarization can be judged upon from the normalized Stokes parameter $S_1$ distribution in the Fourier plane, reaching 0.79 for X- and 0.82 for Y-linear polarizations at the corresponding intensity maxima (Fig. 5d).

## Discussion
The experimental characterization reveals the metasurface performance being worse than expected from the numerical simulations, a deterioration that is understandable for several reasons. First, the simulations were performed for monochromatic (602 nm) fields, whereas optical characterization involved the use of a 16-nm-wide bandpass filter. Second, a GeV-ND is perfectly located with respect to the metasurface in the design, while the experimental precision of ND positioning is limited to 50 nm. Additionally, the radiating dipole is oriented perfectly perpendicular to the surface in the design, whereas that of a GeV-ND center selected for the metasurface fabrication might have nonzero projections on the surface plane. Finally, geometrical parameters of fabricated metasurfaces differ, although on the nm-scale, from the optimized ones. All these factors would result in deteriorated performance, such as broadening of the generated

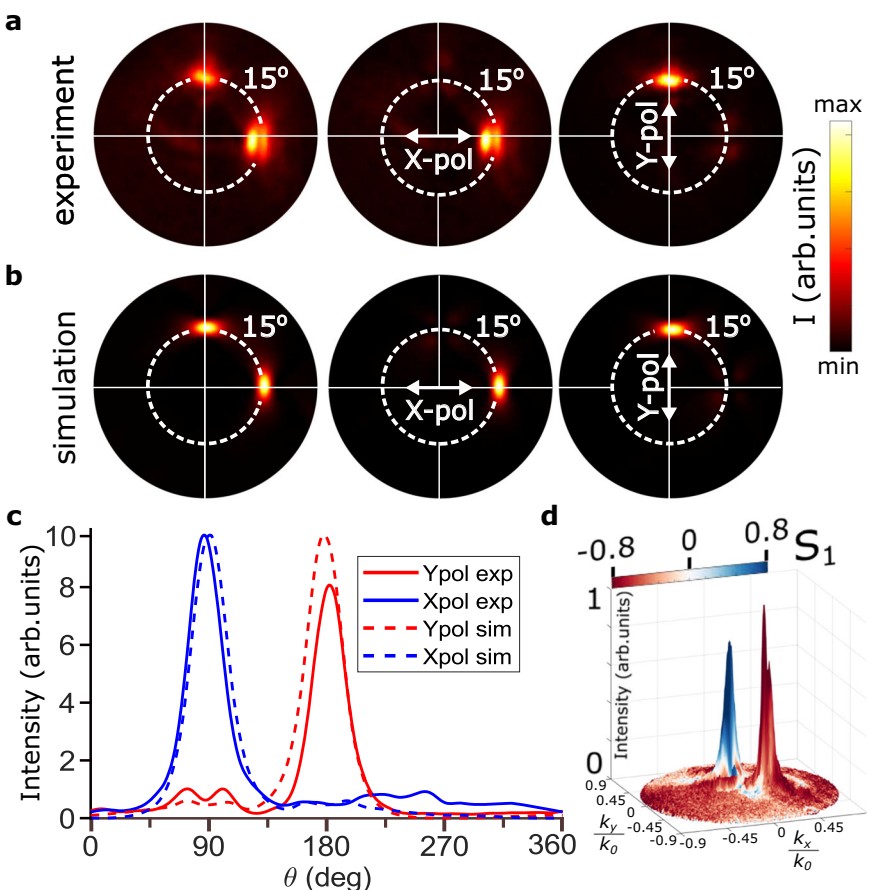

**Fig. 5 | Angular and polarization emission distribution. a** Fourier-plane emission images collected with NA = 0.9 objective from the GeV-ND coupled to the holographic metasurface. From left to right: images taken without analyzer, with analyzer parallel to X-axis, and to Y-axis of the metasurface. **b** Corresponding simulated far-field intensity distributions. Dashed circles mark 15° angular displacements from the normal to the surface. **c** Intensity circular cross section distributions (along the dashed lines) showing clear separation between X- and Y-polarized emission channels. **d** 3D representation of the superimposed experimental distributions of emission intensity (height) and polarization (colour) represented by stokes parameter S1. The angular coordinates $k_x/k_0$ and $k_y/k_0$ are x- and y- wavevector projections on the image plane normalized to the free space wavevector.

beams, polarization cross talk and other distortions in the emission pattern, in comparison with the results of numerical modeling. At the same time, the developed approach for generating two spatially separated single-photon beams propagating along different directions with orthogonal linear polarizations was found to be sufficiently robust and amenable to different radiation wavelengths. The experiments similar to the described above were successfully conducted with another single GeV-ND center as well as with NDs containing multiple Si vacancy (SiV) centers emitting at 738 nm and coupled to metasurfaces with different diameters (Supplementary Figs. S14 and S15).

To summarize, we have developed the approach to realize the on-chip integrated efficient single-photon sources of well-collimated beams propagating along different off-normal directions and featuring different linear polarizations. Using single GeV-ND centers embedded in holographic dielectric metasurfaces supported by silver substrates, we have demonstrated efficient (63%−obtained using numerical simulations, see Supplementary Fig. S6) generation of two well-collimated (divergences < 6.5°) single-photon beams propagating along different 15° off-normal directions and featuring orthogonal linear polarizations. It should be noted that the efficiency can be further increased by using high-refractive index metasurface ridges to boost their out-of-plane scattering and high-quality monocrystalline metal films to reduce the ohmic losses[38]. Thus, simply by changing the ridge material to titanium dioxide (TiO₂), the

external quantum efficiency is expected to reach the level of 80% (Supplementary Section S3), or even exceeding it as described for the bullseye pattern[28]. Importantly, we have shown that the developed approach is robust and flexible and thus can readily be applied to a wide range of QEs, including quantum dots and individual molecules. We believe that on-chip integrated, highly compact and efficient, multifunctional single-photon components demonstrated in our work convincingly showcase the potential of QE-SPP coupled holographic metasurfaces for a wide range of quantum technology applications.

## Methods
### Device fabrication
Metasurfaces are fabricated in a two-step process of electron beam lithography (EBL) using a scanning electron microscope JOEL JSM-6490LV with an acceleration voltage of 30 keV. The substrate consists of a polished Si wafer coated with 3 nm Ti, followed by 150 nm Ag, another 3 nm Ti, and a 30 nm SiO₂ layer, which protects the silver from atmospheric influence. The Ti is added for better adhesion between the materials. In the first lithography step, 35-nm-thick golden alignment markers are deposited using 200 nm thick PMMA positive resist. Further, a water dispersion of GeV-NDs with an average size of ~100 nm is spin-coated on the sample. The choice of NDs is made using a strongly focused radially polarized excitation beam. This beam produces a strong out-of-plane field component in the focal point and

consequently excites GeV centers with a predominant out-of-plane dipole moment most efficiently. In the second lithography step, 200 nm thick HSQ holographic metasurfaces are deposited around individual GeV-NDs with alignment relative to the golden markers (Supplementary Figs. S11 and S12). HSQ is a negative resist. Precise fabrication parameters can be found in Supplementary Section S5.

## Experimental optical characterization

All measurements are performed at room temperature under 532 nm laser excitation. The experimental setup scheme is presented in Supplementary Fig. S13. The excitation laser beam is purified by passing through a single mode polarization maintaining fiber, then converted to radially polarized beam using a Radial Polarization Converter. The Olympus MPLFLN × 100 objective with 0.9 NA focuses the beam onto the sample surface, exciting GeV-ND quantum emitters with a strong normal to surface field component. The optical signal scattered from the metasurface is collected by the same objective. Laser light is filtered out using dichroic mirrors and long-pass filters. Decay-rate measurements are performed using time-correlated single-photon counting technique under weak 10 μW excitation power to keep the GeV in unsaturated emission regime. To purify the emission a narrowband $605 \pm 8$ nm filter was used for all presented optical measurements except spectral ones. For $g^{(2)}$ correlation function measurements 256 ps time-bin width was used. Saturation curves of GeV emission are measured by accumulating counts from both APDs with excitation laser power varying in range from 10 μW to 7 mW. Back focal plane images are captured with a CCD camera, with emission polarization properties measured using a Glan-Thompson polarizer. Exact specifications of the setup components can be found in Supplementary Section S6.

## Pattern generation

The metasurface pattern was generated using Matlab software, calculating the interference between X and Y-polarized beams as signal waves and SPP as the reference wave of a hologram. The pattern was designed considering only in-plane field components on the metal-dielectric interface. The refractive indices of the dielectrics were set to $n(SiO_2) = 1.45$ and $n(HSQ) = 1.41$. The real part of the SPP mode index in the antenna domain was calculated as the average of $N_{eff}(Air) = 1.12$ and $N_{eff}(HSQ) = 1.52$, due to the 50% filling factor. These values correspond to the mode indices of SPP propagation on the Ag–SiO$_2$–Air and Ag–SiO$_2$–HSQ–Air interfaces, respectively. The fixed filling factor allowed to simplify calculations by utilizing the constant SPP mode index and wavelength along the structure. The generation of the pattern is illustrated in Supplementary Section S2.

## Numerical modelling

Numerical simulations were performed using FDTD simulations in a 3D domain. The simulations were conducted at an operating wavelength of 602 nm corresponding to the zero-phonon line (ZPL) of the GeV center. The GeV-ND was modeled as a vertically oriented electric dipole positioned 30 nm above the 30 nm thick SiO$_2$ spacer. The Ag layer had a thickness of 300 nm. The far-field angular distributions were calculated using the built-in Lumerical near-to-far field transformation procedure. The near-field data was collected from the monitor positioned 50 nm above the HSQ structure. Quantum efficiency was calculated by integrating the Poynting vector on a box covering the metasurface, positioned 150 nm above the substrate to avoid encountering the SPP field (Supplementary Figs. S8 and S9). The refractive index of HSQ was set to be $n(HSQ) = 1.41$, extinction coefficient $k = 0$[39]. The optimal dipole separation distance from the SiO$_2$ surface was determined to be between 15 and 80 nm[9]. In the calculation 30 nm separation was used. Optical constants for silver and SiO$_2$ were taken from[40] and[41], respectively.

## Data availability

All data that support the findings of the study are provided in the main text and Supplementary Information file. Raw data are available from the corresponding authors upon reasonable request.

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

## Acknowledgements

Authors gratefully acknowledge Torgom Yezekyan and Volodymyr Zenin for assistance with experimental setup as well as Xujing Liu and Cuo Wu for fruitful discussions on all research steps. S.K. acknowledges financial support from VILLUM FONDEN by a research grant (35950). Y.K. acknowledges the support from European Union's Horizon Europe research and innovation programme under the Marie Skłodowska-Curie Action (Grant agreement No. 101064471). C.M. acknowledges funding from MULTIPLY fellowship under the Marie Sklodowska-Curie COFUND Action (Grant Agreement No. 713694). S.I.B. acknowledges the support from the Villum Kann Rasmussen Foundation (Award in Technical and Natural Sciences 2019).

## Author contributions

S.I.B. conceived the experiment and developed the metasurface design method. D.K. performed numerical optimization, fabrication, and experimental characterization of the device. S.K. assisted in the experiments. C.M. and Y.K. contributed to numerical simulations of device design and optimization. L.F.K., V.A.D. and V.N.A synthesized the GeV nanodiamonds. All authors contributed to discussion and interpretation of the results. D.K. wrote the manuscript with contributions from S.I.B. and S.K. S.I.B. and S.K. supervised the project.

## Competing interests

The authors declare no competing interests.
