## [Peer Review File · Nature Communications]

REVIEWER COMMENTS

Reviewer #1 (Remarks to the Author):

Komisar et al. experimentally demonstrate a metasurface for collimating and polarization filtering the radiation of a single-photon emitter (a Ge vacancy center in a nanodiamond). The design is based on polarization-sensitive coupling of photons to surface plasmon polaritons, and cleverly uses a superposition of two ridge patterns: one for the horizontal polarization and one for the vertical polarization. As a result, single photons are emitted by the quantum emitter in two well-defined and well-separated directions, both at 15° to the metasurface normal, and can be easily collected. The authors also report 63% efficiency of single-photon generation and promise that, with optimized design, 80% can be obtained.

I find this work very useful, because the collection of single photons from nanoemitters, and especially their sorting, is a long-standing goal in quantum nanophotonics. Moreover, the method of ‘a holographic metasurface’, as the authors call it (I prefer to call it a superposition of patterns), has a potential to be implemented in other nanophotonic designs.

However, I see some problems with the presentation, which should be clarified before the paper can be published:

1. In the concluding part, the authors write that they ‘have demonstrated efficient (63%) generation of ... single-photon beams’. It is not clear how the number of 63% is obtained. Usually, an emitter can be characterized by the emission efficiency and collection efficiency. Apparently, the authors mean the latter. But all we can see is how the photon count rate is enhanced (Fig. 4a,c). No absolute efficiency measurement is made. To claim 63% efficiency, additional measurements are needed.

2. Further on Fig. 4c, I notice that it shows the photon emission rate increased by about 10% due to the coupling to the metasurface. Meanwhile, the authors speak of the saturation intensity I_{sat} being increased from 1 Mcps to 1.3 Mcps. This must be the limit to which the count rate tends at a high excitation power. But I cannot see how the solid fitting lines tend to those values. They seem to be at a much smaller distance from each other, 10% or even less. I would appreciate the authors explaining this. Even if this is what the fitting curves show at large P , how accurately does this fit follow from the experimental points?

3. In Fig. 2d, the angular coordinates are not defined.

4. In the Supplementary Information, especially in the experimental part, the English needs correction.

Reviewer #2 (Remarks to the Author):

The authors coupled quantum emitters (QEs), based on single Ge vacancy centers, to surface plasmons originating from nearby metasurface (MS) microstructure. While this is the first study of coupling Ge QEs to metasurfaces I found the approach and results not significant enough to be published in Nature communications. Below I detail my comments.

1. The lack of novelty: There is a similar study published in Nature communications (Nat. Commu. 10, Article number: 2392, 2019) by Bussett group on using QEs, based on single nitrogen vacancy (NV) centers in diamond coupled with Metasurafce lens structure. At least in the later work by Huang et al. a much better enhancement of the quantum properties of QEs in the presence of metasurface lens. See figure 4 where an increase of fluorescence > 200% for the QE-MS coupled vs uncoupled. Also, in another (similar) approach by using of photonic crystal cavity Englund et al. (Nano Lett. 10, 10, 3922–3926, 2010) demonstrated an enhancement of the quantum properties (fluorescence intensity and lifetime) when the NV within the nanodiamond is coupled to the photonics cavity. Strangely neither of the papers are cited in this manuscript.

2. The authors claim an enhancement of the conversion efficiency of QE of 80% depending on the metasurface relaxation. However, in their experimental results (see Figure 4) a barley 30% increase of fluorescence saturation power (1 to 1.3 Mcps) and ~15% decrease of lifetime (from 19 ns to 16 ns).

3. The authors explain the weak enhancement ($\leq 30\%$ observed experimentally) by surface plasmon polaritons (SPPs) scattered into outgoing free-propagating waves by appropriately designed metasurfaces. It is not clear whether this is mainly originated from SPPs or just from the channeling of QE fluorescence due to the increase of NA, as explained in this paper: Huang et al., Nature Communications 10, Article number: 2392 (2019).

Reviewer #3 (Remarks to the Author):

In this manuscript, Komisar et al. experimentally demonstrate holographic metasurfaces coupled to GeV centers in diamonds. Extending to their previous works, they optimized the metasurface structure to maximize the coupling between quantum emitters (QEs) and surface plasmon polaritons (SPPs) supported by the optimized metasurfaces, resulting in efficient separation of two planar polarization. This result is very impressive and novel, and I believe the work is at the level of the expectation in Nature Communications. I have a few questions about the text, and the paper would be better for broad audiences if they could revamp it slightly.

1) One key aspect of the work is the optimization of the metasurface structure, but the article lacks a clear explanation of the optimization techniques used. In general, inverse design methods can involve minimizing a Lagrangian through gradient methods or using neural networks at the cost of high computational power. It would be useful if the authors could provide more details on their optimization approach, particularly for readers who may not be familiar with this field.

2) Minor comment

In addition, the authors could improve the clarity of their figures. For example, in Fig 4b, it would be better to specify the legends, such as $\tau_c = 19$ ns and $\tau_{uc} = 16$ ns, to make it easier for readers to understand the results.

Reviewer #4 (Remarks to the Author):

In this manuscript, the authors involve achieving the simultaneous control of the polarization and direction of single-photon emission using the quantum emitters nonradiative coupled to surface plasmon polaritons. The experiment demonstrates the efficient generation of two well-collimated single-photon beams propagating along different directions with orthogonal linear polarizations.

The research has significant potential for quantum technology applications. However, I suggest this manuscript should be revised to show its novelty, e.g. compare to Sci. Adv. 8, eabk3075 (2022), Sci. Adv. 6, eaba8761 (2020), arXiv:2209.04571. Can this research generate more complex single photon beam with high efficiency or by simpler method?

Response to reviewers

Manuscript ID: NCOMMS-23-09797A

Reviewer #1 (Remarks to the Author):

Komisar et al. experimentally demonstrate a metasurface for collimating and polarization filtering the radiation of a single-photon emitter (a Ge vacancy center in a nanodiamond). The design is based on polarization-sensitive coupling of photons to surface plasmon polaritons, and cleverly uses a superposition of two ridge patterns: one for the horizontal polarization and one for the vertical polarization. As a result, single photons are emitted by the quantum emitter in two well-defined and well-separated directions, both at 15° to the metasurface normal, and can be easily collected. The authors also report 63% efficiency of single-photon generation and promise that, with optimized design, 80% can be obtained.

I find this work very useful, because the collection of single photons from nanoemitters, and especially their sorting, is a long-standing goal in quantum nanophotonics. Moreover, the method of ‘a holographic metasurface’, as the authors call it (I prefer to call it a superposition of patterns), has a potential to be implemented in other nanophotonic designs.

Our response:

We thank the reviewer for appreciating our work. We agree that our approach can be implemented in other nanophotonic designs. We would like to note that we call our metasurfaces ‘holographic metasurfaces’, as we use the holographic method for the generation of metasurface patterns. Please see sections S1 and S2 in the supplementary information (SI).

However, I see some problems with the presentation, which should be clarified before the paper can be published:

1. In the concluding part, the authors write that they ‘have demonstrated efficient (63%) generation of ... single-photon beams’. It is not clear how the number of 63% is obtained. Usually, an emitter can be characterized by the emission efficiency and collection efficiency. Apparently, the authors mean the latter. But all we can see is how the photon count rate is enhanced (Fig. 4a,c). No absolute efficiency measurement is made. To claim 63% efficiency, additional measurements are needed.

Our response:

We thank the reviewer for this comment and agree that more clarifications should be provided. We have now introduced additional details regarding the efficiency in our manuscript and supplementary information (SI) to make it more clear for the readers.

When using the expression “efficient (63%) generation” in the manuscript, we mean the external quantum efficiency η_{QE} defined in our manuscript as “a fraction of the total dipole power radiated into outgoing waves, propagating within a 64° cone (collected by an NA = 0.9 objective)” (page 4).

The experimental measurement of η_{QE} is very complicated, or even impossible because we have access only to the far-field radiation from a quantum emitter (QE) collected by an objective. The nonradiative component of the QE decay η_{nr} cannot be measured. Therefore, we present the theoretically simulated (external) quantum efficiency, η_{QE} , and collection efficiency, η_{CE} , and experimental measurements of the relative change of the collected count rate and lifetime after the QE coupling to the metasurface. We kept all the excitation and measurement conditions the same with only the sample being modified (by fabricating the metasurface around the QE).

In the SI, we discuss in detail the quantum efficiency of metasurface-coupled QEs in section S2.3. To refer the readers to this discussion, we have elaborated in the main text as follows (p. 9, revised manuscript):

“have demonstrated efficient (63% - obtained using numerical simulations, see Supplementary Sections S2.3 and S3) generation of ... single-photon beams”.

2. Further on Fig. 4c, I notice that it shows the photon emission rate increased by about 10% due to the coupling to the metasurface. Meanwhile, the authors speak of the saturation intensity I_{sat} being increased from 1 Mcps to 1.3 Mcps. This must be the limit to which the count rate tends at a high excitation power. But I cannot see how the solid fitting lines tend to those values. They seem to be at a much smaller distance from each other, 10% or even less. I would appreciate the authors explaining this. Even if this is what the fitting curves show at large P , how accurately does this fit follow from the experimental points?

Our response:

We are grateful to the reviewer for this comment. We double-checked the data fitting and found an unfortunate mistake in one of the fitted curve. The experimentally measured dependence of the fluorescence count rate on the excitation laser power $I(P)$ was fitted to the following function:

$$I = \frac{I_{sat}P}{P + P_{sat}},$$

which was used to estimate the value of the saturated count rate and corresponding excitation power. As the reviewer noted, the value $I_{sat} = 1.3$ Mcps does not correspond to experimental data and is wrong. The correct results are: $I_{sat} = 1.0 \pm 0.08$ Mcps; $P_{sat} = 1.8 \pm 0.4$ mW before coupling to the metasurface and $I_{sat} = 1.12 \pm 0.05$ Mcps; $P_{sat} = 1.3 \pm 0.2$ mW for the coupled GeV-ND. Consequently, a count rate enhancement of 12% is observed. Errors amount to 95% confidence intervals as obtained from fitting the data. The experimental error in measurements is less than 1% and lies within the black data crosses in Figure4(c). High-precision experimental measurements were obtained using automatic data acquisition, with >50 data points per measurement.

We have added error intervals to the saturation plot in Figure 4(c), and corrected the corresponding sentences as follows (p. 7, revised manuscript):

“The experimentally measured data was fitted to $I = I_{sat}P/(P + P_{sat})$, where I and P are fluorescence count rate and excitation power, respectively, whereas I_{sat} and P_{sat} are the fitting parameters. It is seen that the saturated emission rate before coupling to the metasurface, $I_{sat(uc)} = 1 \pm 0.08$ Mcps and $P_{sat(uc)} = 1.8 \pm 0.04$ mW, increased after the coupling to $I_{sat(c)} = 1.12 \pm 0.05$ Mcps and $P_{sat(c)} = 1.3 \pm 0.2$ mW.”

3. In Fig. 2d, the angular coordinates are not defined.

Our response:

We thank the reviewer and appreciate this comment. We have defined angular coordinates as $\frac{k_x}{k_0}$ and $\frac{k_y}{k_0}$, x- and y- wavevector projections onto the image plane normalized to wavenumber of light $k_0 = \frac{2\pi}{\lambda_0}$, $\lambda_0 = 602$ nm. Changes are applied to Figures 2d and 5d.

4. In the Supplementary Information, especially in the experimental part, the English needs correction.

Our response:

We are grateful for this important note. We have checked and corrected English in the Supplementary information.

Reviewer #2 (Remarks to the Author):

The authors coupled quantum emitters (QEs), based on single Ge vacancy centers, to surface plasmons originating from nearby metasurface (MS) microstructure. While this is the first study of coupling Ge QEs to metasurfaces I found the approach and results not significant enough to be published in Nature communications. Below I detail my comments.

Our response:

We appreciate the concern expressed, but we have the impression that the reviewer misunderstood some parts in our manuscript, an impression that originates from carefully reading the reviewer's detailed comments. Below, we respond to all comments and hope that the reviewer will now better realize the significance of our work.

1. The lack of novelty: There is a similar study published in Nature communications (Nat. Commu. 10, Article number: 2392, 2019) by Bussett group on using QEs, based on single nitrogen vacancy (NV) centers in diamond coupled with Metasurface lens structure. At least in the later work by Huang et al. a much better enhancement of the quantum properties of QEs in the presence of metasurface lens. See figure 4 where an increase of fluorescence > 200% for the QE-MS coupled vs uncoupled. Also, in another (similar) approach by using of photonic crystal cavity Englund et al. (Nano Lett. 10, 10, 3922–3926, 2010) demonstrated an enhancement of the quantum properties (fluorescence intensity and lifetime) when the NV within the nanodiamond is coupled to the photonics cavity. Strangely neither of the papers are cited in this manuscript.

Our response:

We have very carefully considered the above comments and read the articles the reviewer provided as well as other similar papers. Although the reported studies might look similar at the first glance, the reported functionalities, configurations, and underlying physics are all different compared to those reported in our work. Thus, metalenses that have recently been introduced to manipulate nonclassical light [Nat. Commun. 2019, 10, 2392., Nat. Nanotechnol. 2020, 15, 125., Sci. Adv. 2020, 6, eaba8761.] are still external components, i.e., separated from the photon sources. Essentially, these components operate like their bulky counterparts, influencing the radiation focusing, direction, and polarizations after the quantum emission was generated. Here, we use quantum emitters (QEs) that are non-radiatively coupled to metasurfaces directly forming the QE emission.

We believe that this remarkable feature is sufficiently clearly expressed in our Introduction: “Highly compact and efficient planar photonic components demonstrated in our work represent essentially single-photon sources, collimators and polarizing beam splitters integrated together in on-chip devices...” Note that the differences between using metalenses to manipulate quantum emission and using quantum emitters non-radiatively coupled to metasurfaces have been explicitly, and in detail, compared and summarized in our recent review paper [Adv. Opt. Mater. 2023, 11, 2202759].

As for the reference (Nano Lett. 10, 10, 3922–3926, 2010), we should emphasize that the reported configuration is irrelevant to that presented in our work. Just like the reviewer wrote, it uses a photonic crystal cavity to enhance the emission rate, while our work is mainly about using holography metasurfaces to control the polarization and direction of quantum emission. It is therefore very difficult to find the common ground for their comparison.

In the revision, we have added the following sentence on using metalenses to mold the quantum emission (p. 2, revised manuscript):

“In recent years, metalenses have been introduced to manipulate nonclassical light in a manner of their bulky counterparts, i.e., operating as external components that influence the radiation focusing, direction, and polarizations after the quantum emission was generated [Nat. Commun. 2019, 10, 2392., Nat. Nanotechnol. 2020, 15, 125., Sci. Adv. 2020, 6, eaba8761., Adv. Optical Mater.2023, 11, 2202759].”

2. The authors claim an enhancement of the conversion efficiency of QE of 80% depending on the metasurface relaxation. However, in their experimental results (see Figure 4) a barely 30% increase of fluorescence saturation power (1 to 1.3 Mcps) and ~15% decrease of lifetime (from 19 ns to 16 ns).

Our response:

We believe that this comment is a result of misunderstanding and clarify the corresponding matter below:

We should emphasize that we do not claim “an enhancement of the conversion efficiency of QE of 80% depending on the metasurface relaxation.” Rather, we claim, based on simulations, an external quantum efficiency (not its enhancement) of up to 80% depending on metasurface realization. For the details related to the quantum efficiency in question, we refer to our response to Reviewer #1, comment 1.

In the main text, to avoid misunderstanding, we have modified the corresponding sentence as follows (p. 3, revised manuscript):

“We find that the **external** quantum efficiency of **metasurface-coupled QE emitting photons into the two beams** can exceed 80%, depending on the metasurface realization.”

3. The authors explain the weak enhancement ($\leq 30\%$ observed experimentally) by surface plasmon polaritons (SPPs) scattered into outgoing free-propagating waves by appropriately designed metasurfaces. It is not clear whether this is mainly originated from SPPs or just from the channeling of QE fluorescence due to the increase of NA, as explained in this paper: Huang et al., Nature Communications 10, Article number: 2392 (2019).

Our response:

We would like to point out that, in the provided reference (Huang et al., Nature Communications 10, Article number: 2392 (2019)), the non-plasmonic metalens placed at a focal distance $20\mu\text{m}$ aimed to collect emission from the NV-center located inside a diamond. The well collimated QE emission observed in our experiments is due to appropriately designed metasurfaces that scatter QE-excited SPPs out of the surface plane as also explained in our manuscript. We consider the physical mechanism at work in our case as being completely different from that discussed in the reference provided.

Reviewer #3 (Remarks to the Author):

In this manuscript, Komisar et al. experimentally demonstrate holographic metasurfaces coupled to GeV centers in diamonds. Extending to their previous works, they optimized the metasurface structure to maximize the coupling between quantum emitters (QEs) and surface plasmon polaritons (SPPs) supported by the optimized metasurfaces, resulting in efficient separation of two planar polarization. This result is very impressive and novel, and I believe the work is at the level of the expectation in Nature Communications. I have a few questions about the text, and the paper would be better for broad audiences if they could revamp it slightly.

Our response:

We thank the reviewer for positive assessing our work and for the constructive comments aiming to improve our manuscript.

1) One key aspect of the work is the optimization of the metasurface structure, but the article lacks a clear explanation of the optimization techniques used. In general, inverse design methods can involve minimizing

a Lagrangian through gradient methods or using neural networks at the cost of high computational power. It would be useful if the authors could provide more details on their optimization approach, particularly for readers who may not be familiar with this field.

Our response:

We agree with the reviewer and have now described in detail the procedure of the metasurface pattern generation and efficiency calculation, as well as structure optimization we have used, extending thereby significantly Supplementary Information (SI). Please see the revised sections S1 – S4 in SI.

2) Minor comment

In addition, the authors could improve the clarity of their figures. For example, in Fig 4b, it would be better to specify the legends, such as $\tau_c = 19$ ns and $\tau_{uc} = 16$ ns, to make it easier for readers to understand the results.

Our response:

We appreciate the comment understanding the expressed concern. We have, in the revised manuscript, specified legends in Fig. 4b, and have made other changes to Figures, for example adding error bars to fitted curves in Fig. 4c.

Reviewer #4 (Remarks to the Author):

In this manuscript, the authors involve achieving the simultaneous control of the polarization and direction of single-photon emission using the quantum emitters nonradiative coupled to surface plasmon polaritons. The experiment demonstrates the efficient generation of two well-collimated single-photon beams propagating along different directions with orthogonal linear polarizations.

Our response:

We thank the reviewer for appreciating our work and for the overall positive comments.

The research has significant potential for quantum technology applications. However, I suggest this manuscript should be revised to show its novelty, e.g. compare to Sci. Adv. 8, eabk3075 (2022), Sci. Adv. 6, eaba8761 (2020), arXiv:2209.04571. Can this research generate more complex single photon beam with high efficiency or by simpler method?

Our response:

We would like to first emphasize the main novelty of our paper. The proposed holographic metasurface allows to combine polarization beam splitter, collimator and single photon source in a flat (200 nm thick) on-chip device. Splitting linear polarizations and sending them in different predefined directions with on-chip operation has not been reported before, including in the papers mentioned by the reviewer. In the papers (Sci. Adv. 6, eaba8761 (2020), arXiv:2209.04571) metasurfaces collect the free-space radiation from the QE, and are therefore external components with respect to the QE, being separated from the QE by > 2 μ m distance. Our plasmonic metasurface allows to reduce the thickness of the device to 200 nm. In our previous paper (Sci. Adv. 8, eabk3075 (2022)), the orthogonal polarizations are not separated with respect to propagation directions.

Generally, there are two different routes for manipulating photon emission from QEs, viz. far- and near-field approaches. In the far-field configurations, the photon emission is manipulated using bulky optical components, like lens, mirrors, and polarizers. Metalenses are recently proposed thin optical components for manipulating free space light including quantum emission, but these are still external components for quantum emitters, see details in our response to Reviewer #2 Comment 1. In near-field configurations, like the design here in our work, QEs are non-radiatively coupled to metasurfaces, which can generate polarized and directional photon emission directly by shaping the out-of-plane scattering of surface (non-radiative) modes. This approach is apparently more efficient and compact by virtue of dispensing with the external components.

In the revision, we have described these differences (p. 2-3, revised manuscript):

“Generally, there are two different routes for manipulating photon emission from QEs, viz. far- and near-field approaches. In the far-field configurations, the photon emission is manipulated using bulky optical components, like lens, mirrors, and polarizers. In recent years, metalenses have been introduced to manipulate nonclassical light in a manner of their bulky counterparts, i.e., operating as external components that influence the radiation focusing, direction, and polarizations after the quantum emission was generated [Nat. Commun. 2019, 10, 2392., Nat. Nanotechnol. 2020, 15, 125., Sci. Adv. 2020, 6, eaba8761., Adv. Optical Mater.2023, 11, 2202759]. In near-field configurations, QEs are non-radiatively coupled to metasurfaces, which can generate polarized and directional photon emission directly by shaping the out-of-plane scattering of surface (non-radiative) modes. This approach is apparently more efficient and compact by virtue of dispensing with the external components [Sci. Adv. 8, eabk3075 (2022), Adv. Opt. Mater.2023, 11, 2202759].”

“The approach developed in this work makes one step further by splitting orthogonal polarizations and sending collimated beams in two different directions.”

With respect to generating more complex single-photon beams, we note that our previous work [Sci. Adv. 8, eabk3075 (2022)] mainly focuses on the generation of circular polarized photon emission with OAMs, which occurs along the normal to the surface direction. The developed holography method here can further be applied to generate complex quantum emission by changing the signal waves to corresponding polarizations and directions, which is exactly what we are still working on.

REVIEWERS' COMMENTS

Reviewer #1 (Remarks to the Author):

In the response, the authors clearly answered my questions and I have no other criticism to the presentation. The claim of 63% efficiency was properly softened by explaining that it results from simulations. The other small flaws were corrected as well. I think the paper can be published now.

Reviewer #2 (Remarks to the Author):

The authors replied well to my questions/comments and clarified the novelty of their results. The modified paper based on the input from other reviewers is well integrated in the updated manuscript. I support the paper to be published in Nature Communications after the reviewers clarify this comment:

In their answer to one of my comments they stated: "Here, we use quantum emitters (QEs) that are non-radiatively coupled to metasurfaces directly forming the QE emission" ... and at the same time they replied to Reviewer 1 like "The nonradiative component of the QE decay η_{nr} cannot be measured".

I suggest the authors to clarify this contradiction and update the manuscript accordingly.

Reviewer #3 (Remarks to the Author):

The author substantially revamped their manuscript and addressed the concerns and questions raised by the referees. I recommend this work published in Nature Communications.

Reviewer #4 (Remarks to the Author):

The authors have answered my question about the novelty of their work. The manuscript now is suitable to be published in Nature Comm.

Response to the Reviewers' Comments

Manuscript ID: NCOMMS-23-09797A

Reviewer #2 (Remarks to the Author):

The authors replied well to my questions/comments and clarified the novelty of their results. The modified paper based on the input from other reviewers is well integrated in the updated manuscript. I support the paper to be published in Nature Communications after the reviewers clarify this comment:

In their answer to one of my comments they stated: "Here, we use quantum emitters (QEs) that are non-radiatively coupled to metasurfaces directly forming the QE emission" ... and at the same time they replied to Reviewer 1 like "The nonradiative component of the QE decay η_{nr} cannot be measured".

I suggest the authors to clarify this contradiction and update the manuscript accordingly.

Our response:

We thank the reviewer for supporting publication of our manuscript.

The feeling of contradiction appears because in our response to Reviewer#1 and Reviewer#2, different concepts were explained. We clarify it below:

In the answer to Reviewer#2 "Here, we use quantum emitters (QEs) that are non-radiatively coupled to metasurfaces directly forming the QE emission" we discuss the non-radiative coupling of QE to the metasurface through SPPs. Subsequently, the metasurface scatters SPPs into photons, that propagate to the far-field and are detected.

In the answer to Reviewer#1 "The nonradiative component of the QE decay η_{nr} cannot be measured" we discuss the measurement of nonradiative component in terms of losses. SPPs, that are non-radiatively generated, are coupled to the metasurface and only a certain part of it is scattered and part of it is lost. We cannot measure the non-radiative component in total, as we have access to the part that is radiated to the far-field. As we also wrote in the response to reviewer#1 – "The experimental measurement of η_{QE} is very complicated, or even impossible because we have access only to the far-field radiation from a quantum emitter (QE) collected by an objective.

In the manuscript, all these ideas are sequentially and correctly described, and we do not see any contradictions. It might have happened as the reviewer read responses to different reviewers, and we hope it is clarified now. In this respect, no additions or changes are made in the manuscript.